# High-speed volumetric two-photon fluorescence imaging of neurovascular dynamics

Jiang Lan Fan [1,2], Jose A. Rivera[3], Wei Sun[4], John Peterson[4], Henry Haeberle[4], Sam Rubin[4,8] & Na Ji [3,5,6,7✉]

Understanding the structure and function of vasculature in the brain requires us to monitor distributed hemodynamics at high spatial and temporal resolution in three-dimensional (3D) volumes in vivo. Currently, a volumetric vasculature imaging method with sub-capillary spatial resolution and blood flow-resolving speed is lacking. Here, using two-photon laser scanning microscopy (TPLSM) with an axially extended Bessel focus, we capture volumetric hemodynamics in the awake mouse brain at a spatiotemporal resolution sufficient for measuring capillary size and blood flow. With Bessel TPLSM, the fluorescence signal of a vessel becomes proportional to its size, which enables convenient intensity-based analysis of vessel dilation and constriction dynamics in large volumes. We observe entrainment of vasodilation and vasoconstriction with pupil diameter and measure 3D blood flow at 99 volumes/second. Demonstrating high-throughput monitoring of hemodynamics in the awake brain, we expect Bessel TPLSM to make broad impacts on neurovasculature research.

[1] University of California, Berkeley, CA, USA. [2] University of California, San Francisco, CA, USA. [3] Department of Physics, University of California, Berkeley, CA, USA. [4] Thorlabs Imaging Systems, Sterling, VA, USA. [5] Department of Molecular and Cell Biology, University of California, Berkeley, CA, USA. [6] Helen Wills Neuroscience Institute, University of California, Berkeley, CA, USA. [7] Molecular Biophysics and Integrated Bioimaging Division, Lawrence Berkeley National Laboratory, Berkeley, CA, USA. [8] Present address: LightPath Technologies Inc., Orlando, FL, USA. ✉email: jina@berkeley.edu

Studying hemodynamics in the brain at capillary resolution provides insight into how neural and glial activities are coupled with local metabolism[1–3]. Local changes in brain hemodynamics form the basis of cerebral blood volume (CBV), cerebral blood flow, and blood-oxygenation level-dependent (BOLD) signals in functional magnetic resonance imaging (fMRI)[4,5]. These signals are often used in the medical imaging field as a proxy for neural activity when performing functional experiments or structural targeting[6,7]. Furthermore, abnormal hemodynamics are associated with numerous medical conditions including stroke[8–10], diabetes[11], chronic stress[12], and Alzheimer's disease[13–15] models. Despite the importance of understanding the detailed functional roles of neurovasculature in basic research in order to guide clinical practice, existing optical techniques for interrogating hemodynamics in animals, primarily intrinsic signal optical imaging (ISOI)[7,16–18] and two-photon fluorescence laser scanning microscopy (TPLSM)[19], are limited in scope. ISOI does not require exogenous contrast agents but lacks capillary-resolving spatial resolution and the ability to image at depth. TPLSM can interrogate hemodynamics at high spatial resolution and at depth but is prohibitively slow at imaging brain volumes. An ideal imaging technique should have high spatial resolution to resolve the smallest capillaries and fast volumetric imaging rate to resolve blood flow dynamics.

TPLSM is a popular method for in vivo imaging of the brain because of its ability to image structures at high spatial resolution within scattering tissue[20]. Two-photon excitation requires high peak intensity and thus restricts the fluorescence generation to a thin section at the focal plane, enabling TPLSM to optically section three-dimensional (3D) samples. With high spatial resolution in 3D, it has been routinely used to image neurovasculature at hundreds of microns below the brain surface using fluorescent dyes labeling the blood stream[19,21]. Several studies also demonstrated concurrent imaging of vasculature with neural activity and/or glia using multi-color fluorescent labeling[10,22–26].

To capture hemodynamic events occurring on second (vasodilation and vasoconstriction) to millisecond (red blood cell flow speed) timescales, TPLSM is most often used to capture two-dimensional (2D) frames at video rate (e.g., 30 Hz) or one-dimensional (1D) line scans at hundreds to thousands of lines per second. When applied to monitoring 3D vasculature populations, however, the volumetric imaging speed is limited by the need to serially scan the excitation focus in 3D. For example, for a TPLSM equipped with an 8-kHz resonant galvanometer and bi-directional scanning, imaging a 1 mm × 1 mm × 0.1 mm volume at 1 μm³ voxel size would take over 6 s (not including the additional time needed to move the excitation focus along the optical axis), too slow to image most hemodynamic events.

Even for imaging hemodynamics in 1D or 2D, standard TPLSM faces challenges arising from the micron-sized axial dimension of its excitation focus. Even though standard TPLSM can capture long segments of superficial, horizontally oriented pial vessels and measure their blood flow speed, it captures only cross-sections of penetrating vessels and capillaries that constitute most vasculature below cortical surface[27], making it exceedingly difficult to measure their blood flow speed. In addition, axial movements of the brain, even if only microns in amplitude, can shift structures out of the excitation focus, causing TPLSM to image different sample planes which introduces errors in hemodynamic measurements that are difficult to correct.

Recently, TPLSM methods utilizing axially extended Bessel-like foci[28–30] have been developed for high-speed volumetric imaging of neural activity in the brain[31–33]. Scanning a Bessel focus laterally in 2D generates a projected view of a volume defined by the area scanned and the axial extent of the focus, the latter of which can span tens to hundreds of microns. As a result, the 2D frame rate becomes the effective volumetric imaging speed. In this study, we used a commercial TPLSM equipped with a Bessel focus module and applied it to cortical vasculature imaging in the awake mouse brain at high throughput. Imaging vasculature at up to 600 μm below brain surface, we demonstrated hemodynamic measurement over 1.4 mm × 1.4 mm × 0.1 mm volumes at 15 Hz. The axially elongated excitation volume of Bessel foci produced two-photon fluorescence signal that was proportional to vessel size, which enabled us to use fluorescence signal as a convenient measure of vasodilation and vasoconstriction down to capillary resolution. By imaging vasculature volumetrically at 99 Hz, we were also able to measure 3D blood flow speeds from both superficial and deeply penetrating vessels, including capillaries, up to 3 mm/s.

## Results

**A high-speed TPLSM equipped with a Bessel focus module.** A simplified diagram of the imaging system is shown in Fig. 1a. The entire imaging system including the Bessel focus module is commercially available (Thorlabs Bergamo® II series). The Bessel focus module was located in between the excitation laser and the TPLSM main body, which houses an 8-kHz resonant galvanometer–galvanometer pair scanning system. The combination of a rotatable half-wave plate and a polarizing beamsplitter directed the excitation light from the Gaussian beam path (yellow path, Fig. 1a) to the Bessel beam path (red path, Fig. 1a), where it reflected off a spatial light modulator (SLM) with a concentric binary grating pattern that diffracted the excitation light into an annulus. The excitation light was then focused by a lens and spatially filtered by an annular mask, which blocked the non-diffracted light and transmitted the electric field distribution giving rise to the desired axial focus profile[31]. Removing a mirror in the standard (Gaussian) beam path (yellow path, Fig. 1a) allowed the annular illumination to propagate into the microscope. The annular mask was 4f-conjugated to the microscope's galvanometer scanners and subsequently 4f-conjugated to the back focal plane of a 0.8-NA microscope objective. We characterized the point spread functions (PSFs) of the imaging system in both Gaussian and Bessel modes with sub-diffraction-sized 0.2-μm-diameter fluorescent beads (Fig. 1b). The PSF of the 0.8-NA Gaussian focus had a lateral full-width at half-maximum (FWHM) of 0.67 μm and an axial FWHM of 3.1 μm. The PSF of the 0.4-NA Bessel focus had a lateral FWHM of 0.65 μm, a comparable lateral resolution to the 0.8-NA Gaussian focus due to annular illumination giving rise to a sharper central peak[34], and an extended axial FWHM of 67 μm. The 22× increase in the axial excitation range allowed us to use 2D scanning of the Bessel focus to image a volume that would have required tens of 2D frames captured at different axial positions with the Gaussian focus (Fig. 1c), thereby substantially increasing our imaging throughput.

**Volumetric structural imaging of blood vessels with Bessel focus scanning.** We first performed structural imaging over 1.4 mm × 1.4 mm areas of vasculature in five mice using both Gaussian and Bessel foci, with representative images from one mouse shown in Fig. 2. Gaussian structural imaging was performed on each mouse to obtain 3D morphology of their vasculature network, which enabled us to assign depth information to Bessel data post hoc. Figure 2a–c shows Gaussian TPLSM images of vasculature labeled with dextran-conjugated Texas Red (peak emission: 615 nm) at 55, 225, and 420 μm below the top surface of the dura mater, respectively. Large segments of pial vessels were visualized in the most superficial image (Fig. 2a). Within cortex, penetrating vessels and capillaries showed up

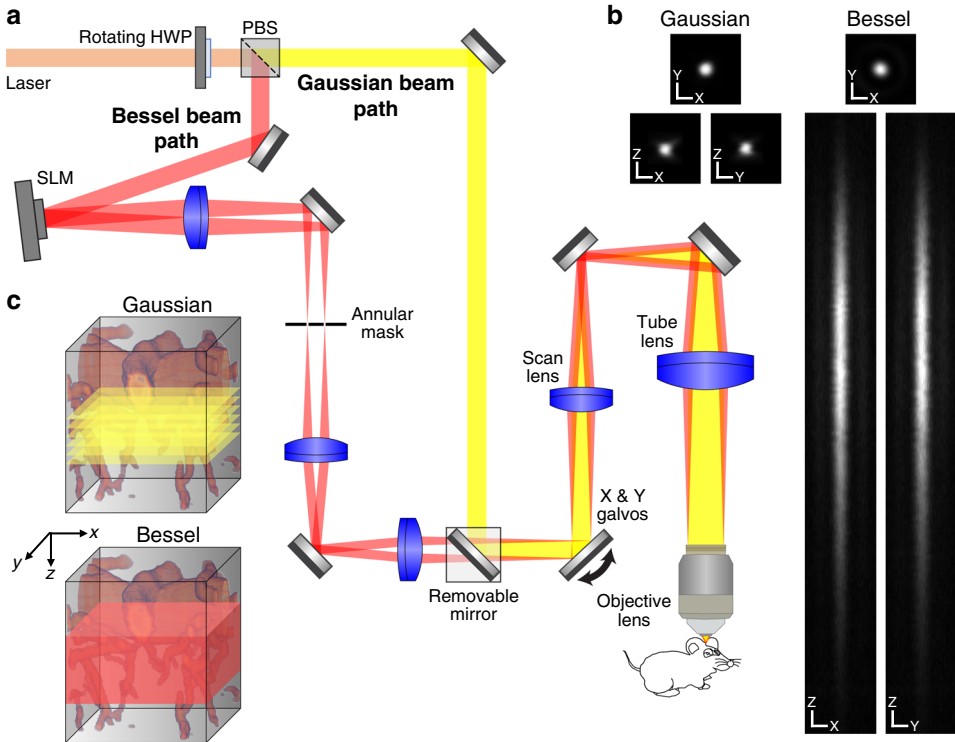

**Fig. 1 Design and characterization of a commercial two-photon laser scanning microscope with a Bessel focus module. a** Schematic of the microscope. A half-wave plate (HWP), a polarizing beamsplitter (PBS), and a removable mirror allow switching between Bessel (red) and Gaussian (yellow) beam paths. In the Bessel path, a spatial light modulator (SLM) and a lens generate an annular illumination pattern, which after spatial filtering by an annular mask is imaged via a 4f system onto the galvos and subsequently imaged via a scan and tube lens pair onto the objective lens back focal plane. **b** Lateral and axial point spread functions for Gaussian and Bessel foci. X and Y scale bars: 1 μm. Z scale bar: 5 μm. Results from one 0.2-μm-diameter bead. **c** Schematic comparison of Gaussian and Bessel volumetric TPLSM methods. Gaussian volumetric imaging requires multiple 2D frames taken at different Z-positions, while Bessel volumetric imaging is achieved with a single frame.

sparsely because only their cross-sections were imaged by the tightly axially confined Gaussian focus. To capture volumes of vasculature, we first scanned the Gaussian focus in 3D. Figure 2d–f shows Gaussian image stacks of vasculature within 0–110, 170–280, and 370–470 μm depth ranges, respectively, taken at 1-μm z-steps and color-coded by depth. Consistent with the Gaussian images in Fig. 2a–c, we observed horizontally located dural and pial vessels in the superficial brain (dark blue and blue-green vessels in Fig. 2d, respectively). Within the cortex, the vasculature was dominated by capillaries that extended over large depth ranges (segments color-coded from blue to red, Fig. 2d–f and insets). Next, by scanning the axially extended Bessel focus (Fig. 1b) in 2D, we imaged 100–110-μm-thick volumes of vasculature simultaneously (Fig. 2g–i) with sufficiently high lateral resolution to resolve individual capillaries (cf. insets in Fig. 2d–i), indicating that Bessel focus scanning drastically improved volumetric imaging throughput without compromising lateral resolution (Supplementary Movie 1). Penetrating arterioles or venules that ran parallel to the Bessel focus (arrow heads, Fig. 2e, f) generated the brightest fluorescence signal (arrow heads, Fig. 2h, i) because they contained the most fluorophores to be excited by the axially extended Bessel focus. As an example of multicolor volumetric imaging, we simultaneously imaged vasculature and glia cells expressing green fluorescence protein (GFP) in the same cortical volume (i.e., data in Fig. 2a and j, b and k, g and l were acquired simultaneously). A 2D Bessel focus scan recorded vasculature and glia in the 110-μm-thick volume, with the low density of the glia allowing individual cells to be separately identified in the projected view of the Bessel image (Fig. 2j–l and insets).

**Fluorescence brightness of vasculature in Bessel images is positively correlated with vasculature size**. We systematically compared the vessel and capillary sizes measured by Bessel versus Gaussian focus scanning. We selected 60 segments in Fig. 3a (1.4 mm × 1.4 mm × 100 μm volume, 0–100 μm below the top surface of the pia mater, 1024 × 1024 pixels) and compared their sizes with those measured from the corresponding 3D Gaussian stack (Supplementary Fig. 1, Methods). We found that the sizes obtained from both datasets were highly correlated ($R^2 = 0.88$), which indicated that the projected structural images of vasculature volumes from Bessel focus scanning provided accurate measurements of vessel sizes.

Interestingly, visual inspection of vessels and capillaries in Gaussian and Bessel images (Fig. 3b) revealed distinct trends: in Gaussian images, in-focus vessels of different diameters had similar fluorescence brightness, while in Bessel images, larger vessels were much brighter than fine capillaries (Fig. 3b, arrows). These trends were confirmed by systematic analysis on size-brightness correlations using the same 60 vessel segments in Supplementary Fig. 1. We observed weak positive correlation between fluorescence brightness and vessel size in their Gaussian images ($R^2 = 0.08$), but strong positive correlation ($R^2 = 0.75$) in their Bessel image (Fig. 3c). The strong positive correlation between fluorescence brightness in the Bessel image and vessel size is because larger vessels allowed more fluorescent molecules to be excited along the axial direction by the axially extended Bessel focus (67 μm FWHM), leading to larger fluorescence signal. In contrast, the Gaussian focus extended much less axially (3.1 μm FWHM). As a result, its signal did not increase for vessels with larger axial dimensions.

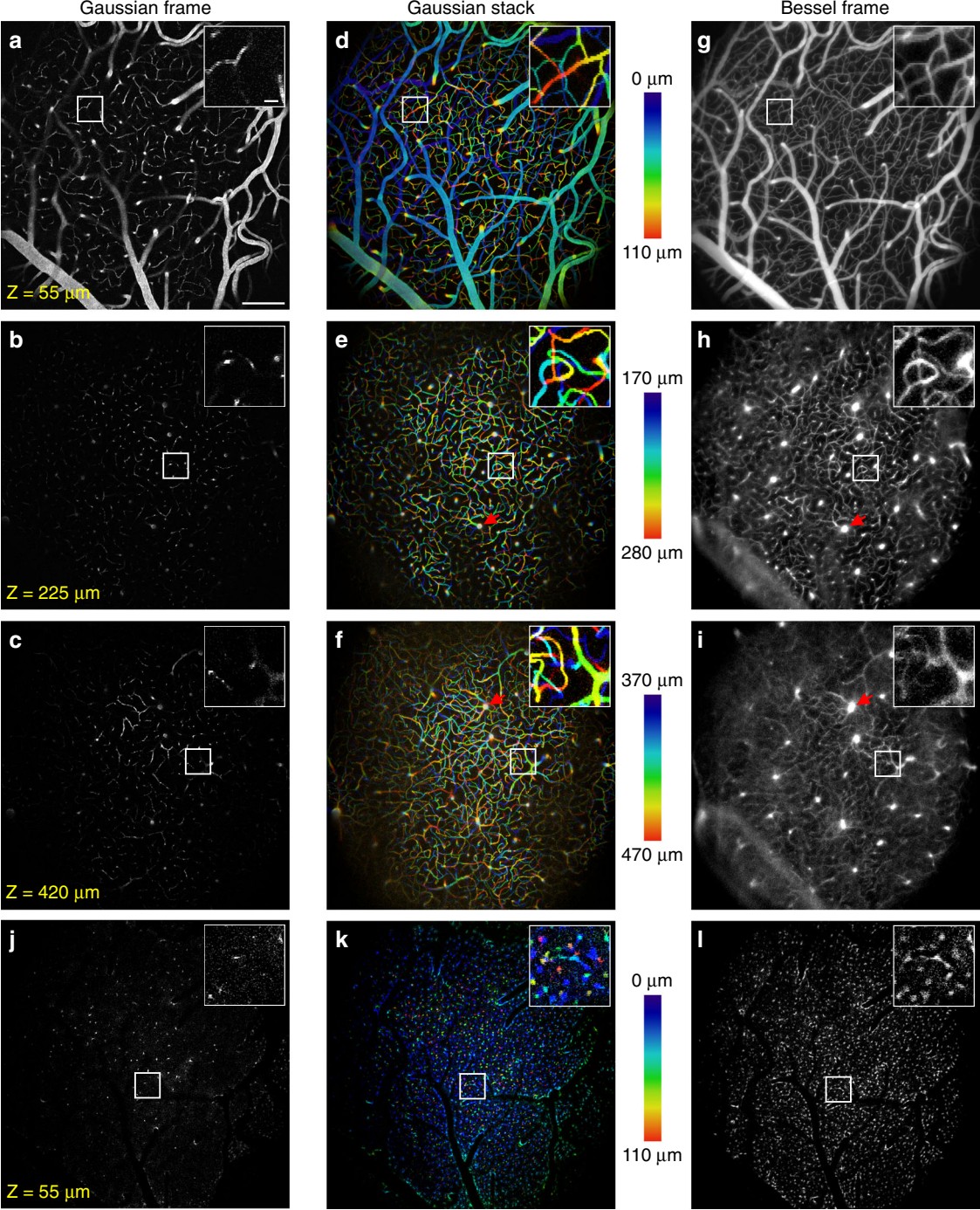

**Fig. 2 In vivo volumetric structural imaging of vasculature and glia with Bessel TPLSM. a–c** Gaussian TPLSM images of vasculature labeled with dextran-conjugated Texas Red at 55 µm, 225 µm, and 420 µm depths, respectively, over a 1.4 mm × 1.4 mm area in the mouse cortex in vivo. **d–f** Gaussian TPLSM image stacks of vasculature at 0–110 µm, 170–280 µm, and 370–470 µm depths, respectively, color-coded by depth. **g–i** Scanning the Bessel focus in 2D captured all vasculature in the volumes within (**d–f**). **j–l** Gaussian (**j**, **k**) and Bessel (**l**) images of GFP-expressing glia imaged concurrently with **a**, **d**, and **g**, respectively. Insets: zoomed-in views of the white-boxed regions. Red arrowheads: vessels oriented parallel to the Bessel focus; **a** and **g** use a grayscale on the normalized square root of fluorescence signal to highlight dim structures without saturating bright structures. All other panels use grayscale on the normalized linear fluorescence signal. Representative data from four mice. Scale bars: 200 µm for full FOV; 20 µm for insets. Post-objective excitation power: Gaussian: 45–177 mW; Bessel: 217 mW.

**High-throughput volumetric imaging of vasodilation and vasoconstriction**. The observed positive correlation between fluorescence brightness and diameter of distinct vessel segments in the Bessel image, although high in value, cannot be relied upon to reach conclusions on relative sizes for different vessels. Even

for the vessels in the same field of view (FOV), variations in local tissue scattering, aberration, and orientation relative to the excitation focus led to deviations away from perfect signal-to-size correlation. However, for the same vessel segment, changes in its fluorescence brightness in Bessel TPLSM should be strongly

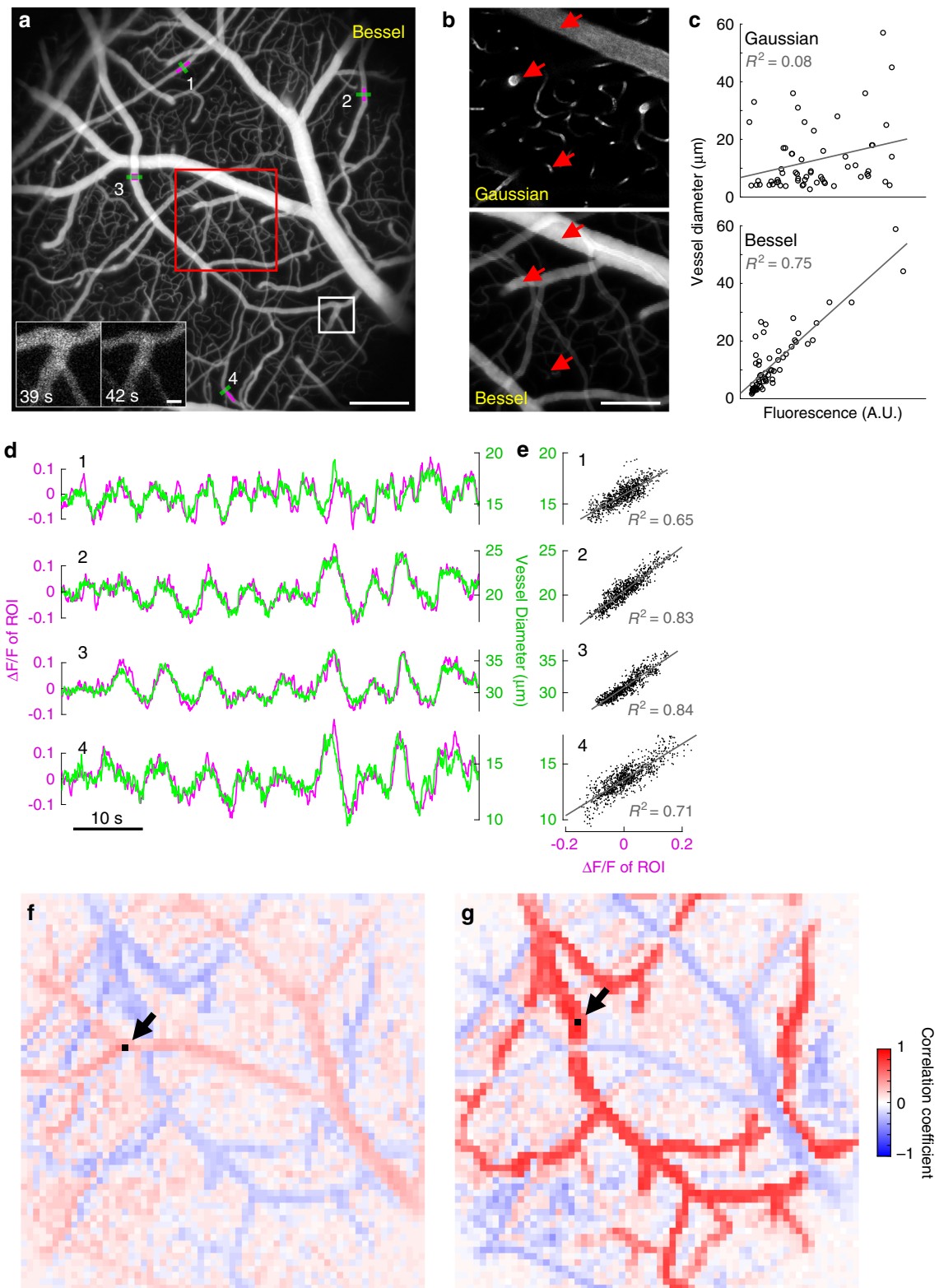

correlated with changes in its size, and thus can be reliably used to detect vasodilation and vasoconstriction. As an example, we used Bessel focus scanning to monitor the vasculature dynamics in 1.4 mm × 1.4 mm × 100 μm volumes at 15 Hz. At this volumetric rate, we captured the hemodynamics of vasodilation and vasoconstriction in five awake mice under fixed ambient lighting condition (e.g., insets in Fig. 3a, Supplementary Movie 2). We

found that temporal variations in vessel size over 1 min of recording (e.g., green traces in Fig. 3d, measured along green lines for four example vessel segments labeled 1–4 in Fig. 3a) were positively correlated with the brightness changes of its lumen (e.g., magenta traces in Fig. 3d, averaged brightness within the magenta regions of interest (ROIs) in Fig. 3a; quantified as ΔF/F, with F being the median value of the time trace) (Fig. 3e,

**Fig. 3 Bessel TPLSM signal is correlated with vessel size and captures distributed dynamics of vasodilation and vasoconstriction in 3D. a** A 1.4 mm × 1.4 mm × 0.1 mm volume of vasculature imaged at 15 Hz using Bessel TPLSM, visualized in grayscale on the normalized square root of fluorescence signal. Insets: zoomed-in views of the white-boxed region at two time points, showing changes in vessel size. **b** Gaussian (single plane at $Z = 50\,\mu m$) and Bessel images of the red-boxed region in **a** captured at different times, visualized in grayscale on the normalized fluorescence signal. Red arrows point to three vessels (large, medium, small) to highlight the differences in their fluorescence signal strength between Gaussian and Bessel TPLSM. **c** Fluorescence vs. vessel diameter for 60 vessel segments in **a** (see Supplementary Fig. 1) imaged with Gaussian or Bessel TPLSM. **d** Time traces of fluorescence signal changes of the magenta regions of interest (ROIs) and blood vessel diameter measured along the green lines in **a** for four vessel segments. **e** Scatter plot of the data in **d**. **f**, **g** Maps of cross-correlation coefficients between ROIs tiling the FOV in **a**, and a reference ROI (indicated by arrows and black squares). Representative data from four mice. Scale bars: 200 μm for **a**, **f**, **g**; 20 μm for insets in **a**; 100 μm for **b**. Post-objective power: Gaussian: 45 mW; Bessel: 217 mW.

Methods). The observed dilatory oscillations in Fig. 3d were similar in frequency (~0.1 Hz) to previously observed cerebral vasomotion[35].

We carried out similar analyses for capillaries (vessels with diameters <6 μm; Supplementary Fig. 2) in two animals, at 30 Hz and 0.2 μm pixel size. In both Gaussian (Supplementary Fig. 2c, d) and Bessel (Supplementary Fig. 2e) modes, we measured diameters of capillary segments and observed dilation and constriction from some capillaries. Our observation that some but not all capillaries dilate is consistent with previous studies where only a fraction of capillaries was found to undergo measurable dilation[36,37]. Those capillaries that did dilate show similar magnitudes of diameter change to those observed previously in response to forepaw or whisker pad stimulations[36–39].

For all example vessels, vessel diameter and fluorescence brightness were highly positively but not perfectly correlated. The lack of perfect correlation was caused by blood cells, which were not labeled by fluorescent dye. Whenever they flowed through the excitation focus, they reduced the fluorescence brightness even when the vessel size stayed constant. To reduce blood-cell-induced fluctuations in fluorescence, we temporally averaged the fluorescence and diameter data (Methods; 0.33 s moving average for non-capillaries and 5 s moving average for capillaries) and observed the fluorescence brightness to closely reflect the vessel diameter.

This relationship between vessel size and fluorescence brightness provided a simple brightness-based method to detect changes in blood vessel size over an entire volume without the need for image segmentation. As an example, we separated the Bessel imaging volume into 64 × 64 ROIs (with 16 × 16 pixels or 21 μm × 21 μm for each ROI) and cross-correlated each ROI's fluorescence time trace to a reference ROI's time trace. For example, with two distinct ROIs as references (arrows, Fig. 3f, g, respectively), the resulting correlation maps highlighted two vasculature populations with positively correlated diameter changes within themselves and negatively correlated diameter changes with each other (Fig. 3f, g). The positive correlations likely arose from the propagation of the dilatory signals along the vessel walls of connected vasculature[26,38,40]. The reference ROI in Fig. 3g had a ΔF/F time trace with a standard deviation 3.6× greater than the reference ROI in Fig. 3f (Supplementary Fig. 3). Therefore, the ROI in Fig. 3g likely belonged to surface arterioles and the ROI in Fig. 3f likely belonged to venules due to the differing dilatory mechanisms between arterioles and venules[3,19,41]. Pixels with positive correlation with the two ROIs in Fig. 3f and g, respectively, revealed surface venule and arteriole populations. Although the discovery of these two distinct populations was expected, anticorrelated dilatory dynamics of two vasculature populations occupying the same brain volume, to our knowledge, has not been reported before.

Next, we investigated how vasodilation and vasoconstriction were entrained with animal arousal level as reflected by the pupil diameter of the mouse eye. We simultaneously recorded a quietly awake mouse's ipsilateral eye under fixed lighting conditions with an infrared camera (15 Hz, 10 min) and a 1.4 mm × 1.4 mm × 100 μm volume of its brain vasculature with Bessel TPLSM imaging (0–100 μm below the top surface of the dura mater, 15 Hz, 10 min) (Fig. 4a, b). We tracked and extracted the pupil profiles (Fig. 4c, yellow ovals) and observed large spontaneous changes in pupil diameter over the time course of seconds (Fig. 4d, top trace). As there was no change in ambient light level, these changes in pupil diameter have been previously shown to reflect the arousal level and/or switching of cortical states[42]. Similar to the analyses performed in Fig. 3f and g, we calculated how fluorescence signal traces of 64 × 64 ROIs were correlated with pupil diameter (Fig. 4e). We observed populations of vasculature whose brightness either positively or negatively entrained to pupil diameter (Supplementary Movie 3). As indicated by two example time traces (ROIs 1 and 2, Fig. 4d, e), fluorescence signal, and thus vessel sizes according to Fig. 3, exhibited changes either positively or negatively correlated with animal arousal. Based on their location and morphology, ROI 1 and ROI 2 were identified to be a surface artery and a dural arteriole, respectively, with distinct mechanisms regulating their dilation[41]. We noticed that pupil dilation events displayed slower rise and decay dynamics when compared with blood vessel dilation events.

**High-speed volumetric measurement of cerebral blood flow.** Finally, we applied Bessel TPLSM volumetric imaging to 3D cerebral blood flow speed measurements in four mice at 99 Hz volumetric rate, an imaging speed fast enough to measure blood flow speeds around 3 mm/s and capture red blood cell (RBC) motion in most capillaries, pial venules, and some pial arterioles[19,21]. As shown for a representative volume (416 μm × 416 μm × 80 μm, 128 × 128 pixels, 0–80 μm below the top surface of the pia mater), Bessel TPLSM was able to record from all vessels and capillaries within the imaging volume (Fig. 5a, Supplementary Movie 4).

We first traced blood vessel segments by hand over the Bessel image (e.g., Fig. 5a, blue and red traces) and then measured their 3D lengths by locating the same segments in skeletonized 3D Gaussian structural data (Fig. 5c, Methods). In total, we traced 63 vessel segments in seven volumes from four animals (49–549 μm in 3D length, with a median length of 174 μm), of which 28 segments had enough spatial extent along Z to require the use of Bessel focus to measure their blood flow speed. In order to measure blood flow speed, pixels along the traced segments in the Bessel image series were plotted against time and hereafter referred to as kymographs (Fig. 5d). Since the horizontal axes of the kymographs represented 2D-projected vessel lengths, non-linear transformations were performed on the horizontal axes such that they represented 3D lengths of the blood vessel segments (Fig. 5e). Because RBCs were not labeled with fluorescence, they showed up as dark streaks in the kymographs.

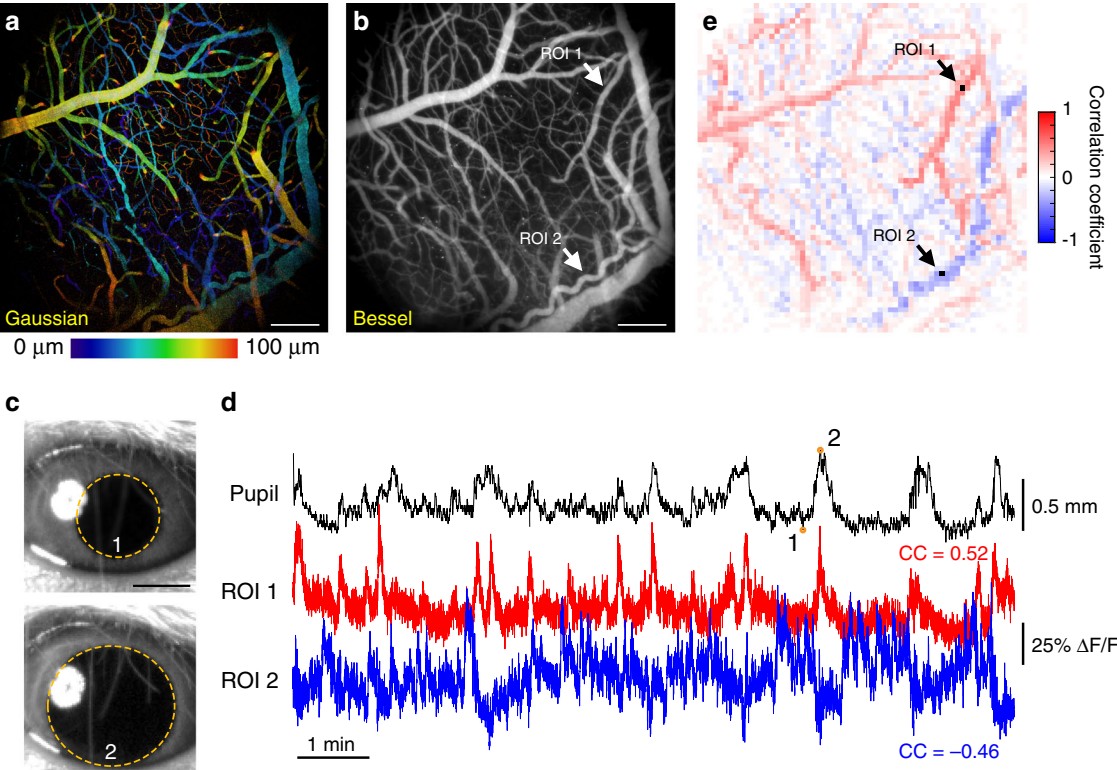

**Fig. 4 Entrainment of vasodilation and vasoconstriction of a 3D vasculature network with pupil diameter measured by Bessel TPLSM. a** A 1.4 mm × 1.4 mm × 0.1 mm volume of vasculature imaged at 15 Hz with Gaussian TPLSM, color-coded by depth. **b** Bessel TPLSM image of the same volume in **a**, visualized in grayscale on the normalized square root of fluorescence signal. **c** Example pupil images at time points 1 and 2 of the ipsilateral eye acquired concurrently with vasculature imaging. Dashed yellow ovals: pupil profiles automatically segmented from video data. **d** Pupil diameter and signal time traces of two regions of interest (ROIs, arrows and black squares in **e**), showing positive and negative correlation with pupil diameter, respectively. CC: correlation coefficients. Orange circles on pupil time trace indicate time points 1 and 2. **e** Map of cross-correlation coefficients between ROIs tiling the FOV in **b** and pupil diameter. Representative data from four mice. Scale bars: 200 μm for **a**, **b**, and **e**; 1 mm for **c**. Post-objective power: Gaussian: 47 mW; Bessel: 167 mW.

From the time required for RBCs to traverse the full length of the segment (i.e., the slopes of the dark streaks in the kymographs), blood flow speed can be calculated. To this end, we applied an automated blood flow speed measurement method based on Sobel filtering and iterative Radon transforms[43] to the kymographs and measured temporally varying blood flow speeds over 1-min-long datasets (yellow lines, Fig. 5e, f). We used the median flow speeds as the representative speeds of the traced vessels (dashed line, Fig. 5f). We overlaid the flow speed on 11 vessels in Fig. 5b to provide a visual representation of flow speeds from the volumetrically distributed blood vessels spanning 80 μm of depth. We plotted the median flow speed against blood vessel diameter for all 63 blood vessel segments and found a positive correlation between vessel diameter and blood flow speed (Fig. 5g), consistent with observations that blood flows faster in larger vessels[8,11,44].

## Discussion

We presented in this study successful fast structural and functional imaging of neurovasculature over large volumes (up to 1.4 mm × 1.4 mm × 110 μm) and at high speed (up to 99 Hz) using a commercially available TPLSM with a Bessel focus module. Scanning the axially extended Bessel focus in 2D enabled us to obtain projected images of 3D volumes, thus drastically increases the volumetric imaging speed and reduces imaging data size. We demonstrated high-throughput tracking of vasodilation, vasoconstriction, and blood flow speed in 3D in the awake mouse cortex.

When Bessel TPLSM was applied to neurovasculature imaging in the mouse cortex, fluorescence intensity proportionally varied with vessel size, thus providing an alternative approach to image segmentation for measuring the dynamics of vessel dilation and constriction. Importantly, this simplified analysis approach combined with the small data size of Bessel TPLSM allowed for fast data analysis using ImageJ and MATLAB without the need for dedicated hardware or optimized software.

By extending the excitation volume axially, a Bessel focus allowed us to measure blood flow in vessels and capillaries that were not parallel to the axis of the Bessel focus. In contrast, in conventional TPLSM imaging with the more axially confined Gaussian focus, many vessels appear only as cross-sections, which makes blood flow measurement difficult. If it is important to measure blood flow from vessels that are parallel with the axis of the Bessel focus, the annular illumination at the back focal plane of the microscope objective can be translated to generate a Bessel focus traveling along a different direction from the original focus[29,45,46] and a distinct projected view of the vasculature.

In contrast to blood vessel cross-sections captured with Gaussian focus, full length images of blood vessels captured with Bessel focus scanning do not suffer from axial motion artifacts because axial motions several microns in amplitude do not shift the structures out of the extended Bessel excitation focus. Lateral motion artifacts were easily correctable because the fast imaging rates afforded by the resonant galvanometer scanning minimized in-frame non-rigid sample motions. Bessel TPLSM's robustness

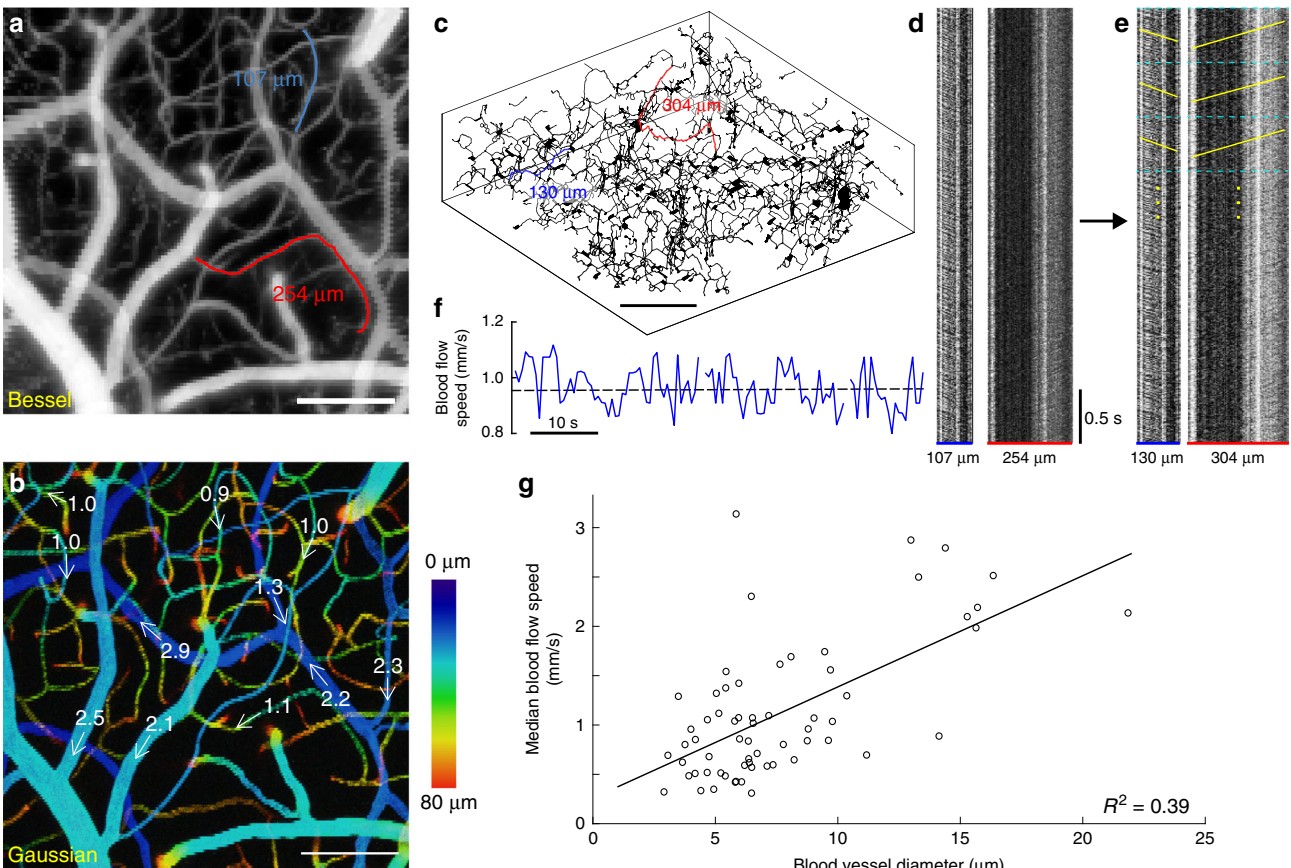

**Fig. 5 High-speed volumetric measurement of cerebral blood flow speed with Bessel TPLSM. a** Bessel image of a 416 μm × 416 μm × 80 μm volume of vasculature that was imaged at 99 Hz with Bessel TPLSM for blood flow speed measurements, with structures visualized in grayscale on the normalized square root of fluorescence signal. Blue and red lines trace two example vessel segments. **b** Gaussian image stack of the same volume in **a**, color-coded by depth and overlaid with median blood flow speeds (in mm/s) of 11 blood vessel segments. **c** Skeletonized 3D Gaussian stack from **b** for the measurements of blood vessel segment 3D lengths. **d** Kymographs of two example vessel segments in **a**, obtained by plotting pixels along the traced blood vessels (horizontal) across time (vertical). Dark diagonal streaks are caused by RBCs traveling along the vessel segment. Note that the horizontal axis represents projected 2D distance. **e** Kymographs of the same vessels after nonlinear transformation so that the horizontal axis represents 3D distance. For flow speed measurement, each kymograph was divided into 0.5-s-long blocks (cyan dashed lines, Methods). Yellow lines represent the distance–time relationship of the blood flow measured for each block. **f** Changes in blood flow speed over 1 min for the blue vessel segment in **a**. **a**–**f**: Representative data from eight volumes in four mice. **g** Median blood flow speed plotted against blood vessel diameter for 63 vessel segments between 0 and 180 μm below dura, collected from four mice. Scale bars: 100 μm for **a**–**c**. Post-objective power: Gaussian: 45 mW; Bessel: 217 mW.

against motion artifacts is critical for studying hemodynamics in awake animals and would be especially important when working with disease models where motor dysfunctions or seizures cause large sample motion.

Because structures at different depths within the excitation volume appear in the same projected images, Bessel TPLSM performs best with sparsely labeled and high-contrast structures, making it suitable for imaging neurovasculature, which are sparse in the brain and typically labeled with bright fluorescent dyes. With the available laser power and dextran-conjugated Texas Red dye, we were able to image neurovasculature up to 800 μm below the brain surface for conventional Gaussian TPLSM and 600 μm for Bessel TPLSM. The reduced imaging depth of Bessel TPLSM was partly due to the higher fraction of excitation energy distribution in the side rings of the Bessel focus[34], leading to less effective two-photon excitation for the same post-objective laser power. The excitation energy in the side rings also reduces the image contrast and lowers the signal-to-background ratio (SBR). To image vasculature structures at depth, more average power needs to be deposited into the mouse brain with Bessel TPLSM

than Gaussian TPLSM. However, the highest average power utilized in this study (up to 217 mW) was under the threshold for heating-induced damage in the mouse brain in vivo[47] and below the average powers that were utilized in other high-speed brain imaging technologies[48,49].

Besides increasing laser power, improvements in neurovasculature imaging depth can be accomplished by using longer-wavelength excitation, because scattering mean free path increases with longer-wavelength photons[50]. For example, indocyanine green is two-photon excitable with 1700 nm light and was used to image vasculature down to 2 mm with Gaussian excitation[51]. Three-photon fluorescence excitation using Bessel foci also employs longer-wavelength excitation light and has the added benefit of suppressed side-ring fluorescence excitation due to the higher non-linear process[52].

Bessel beam imaging depth can also be improved by adaptive optics[53,54], a method which corrects for optical aberrations of the imaging-forming excitation light introduced by the brain tissue. We observed distinct effects of optical aberrations caused by large surface vessels for Gaussian and Bessel foci. Large surface vessels

aberrated the Gaussian focus and led to reduced excitation efficiency and thus dark shadows in the Gaussian image stacks (Fig. 2e, f). In contrast, large surface vessels degraded spatial resolution and signal-to-background contrast in Bessel image stacks (Fig. 2h, i) suggesting that aberrations in the Bessel focus led to asymmetric focal energy distribution but not a decrease in the overall two-photon excitation efficiency. Removing these sample-induced aberrations would improve excitation efficiency and resolution, which in turn would increase SBR and thus imaging depth.

Even without the improvements outlined above, the current implementation already allowed us to apply Bessel focus scanning TPLSM to monitor hemodynamics in 3D at capillary resolution. We observed vasodilation and vasoconstriction in superficial blood vessels[55–57] both directly by measuring vessel diameter changes and indirectly through Bessel focus excited fluorescence signal. We observed size changes in some capillaries, which may be regulated by the flow in nearby arterioles or due to active control by more proximal factors such as pericytes[39,57–59]. We observed two vasculature populations negatively correlated with one another in terms of their dilation/constriction dynamics, possibly distinguishing between arterioles and venules. We also observed entrainment of brain hemodynamics to pupil diameter, a phenomenon that was reported in the past using ISOI[60,61]. Since pupil diameter is closely tied to brain arousal levels[62–64] and cortical states[42,65], this entrainment of vessel and pupil dilations reflects underlying neural activity dynamics including the release of vasoactive neurotransmitters[66]. We noticed faster temporal dynamics in vessel size changes than pupil diameter changes, which to our knowledge have not been directly compared in literature. However, this observation is consistent with previous reports that neurovasculature responds to neural activity on sub-second timescales[17,38] while the pupil responds on the scale of seconds[65].

In addition to vasodilation and vasoconstriction, we also used Bessel focus scanning TPLSM to measure blood flow speeds in 3D at 99 volumes per second from both dural and cortical vessels as well as capillaries. Following the calculations by Shih et al.[19], we estimated the upper bound of RBC flow rate that can be captured at this speed to be 4.3 mm/s, faster than the flow rates of most neurovasculature[19,21]. Compared with previous studies using line scan paths to simultaneously measure blood flow speed in multiple vessel segments which does not scale well with the number of features[67,68], Bessel focus scanning could record from all blood vessels in an imaging volume. To capture even faster blood flow rates if needed, volumetric imaging exceeding 100 Hz can be achieved by using faster optical scanning[69,70], imaging smaller volumes, or sampling at a lower pixel density. Improving SBR by methods described above would become essential when the achievable imaging speed becomes limited by fluorescence signal strength.

Finally, Bessel focus scanning TPLSM enables volumetric imaging of neurovascular dynamics without sacrificing the high lateral resolution and multi-color imaging capability of Gaussian TPLSM. With the developments of bright and efficient two-photon excitable fluorescent sensors for neural and glial activity reporting[71], Bessel TPLSM can simultaneously image neurovasculature with neurons and glia to generate high-throughput, rich datasets for uncovering the detailed dynamics of neurovascular coupling in healthy and diseased brains[55]. Vascular imaging with Bessel TPLSM can also be applied to other vasculature systems such as those in the retina, spinal cord, or skin.

## Methods
### Design and characterization of a commercial TPLSM with a Bessel focus module. 
We designed and constructed a commercially available Thorlabs Bergamo® II multiphoton microscope with a Bessel focus module for high-speed volumetric imaging (Fig. 1a). All hardware controls and data acquisition were performed using the ThorImage software. A titanium–sapphire laser (Chameleon Ultra II, Coherent Inc.) tuned to 920 nm was used as the two-photon excitation source for all experiments. A 16×/0.8-NA water-dipping objective lens (Nikon) was used for all characterization and imaging data in this study. Bessel and Gaussian beam paths were switchable using an integrated, software-controlled rotatable half-wave plate combined with a polarizing beamsplitter and a removable mirror. Compared to the Gaussian beam path, the Bessel beam path had an additional lens (focal length = 400 mm) with a liquid crystal SLM (ODPDM512-1064, Meadowlark Inc.) at its front focal plane and an annular aperture mask at its back focal plane. A 0-π concentric binary phase grating pattern was presented on the SLM so that its 1st order diffraction ring was focused on the annular mask by the lens. The annular mask was custom fabricated (chrome deposited on quartz, Photo Sciences Inc.) to block 0th and higher order diffractions and its parameters (inner diameter = 2.3 mm, outer diameter = 2.5 mm) were chosen to generate a Bessel focus with a theoretical axial full-width at half-maxima (FWHM) of 85 μm at the focal plane of the 16×/0.8-NA objective, as calculated following Lu et al.[31]. The annular mask was 4f-conjugated to the non-resonant galvo surface in the galvo-resonant galvo scanning system. Fluorescence emission was collected by two GaAsP photo-multiplier tubes (PMTs) with emission filters for simultaneous 2-color imaging of green (525/50 nm) and red (607/70 nm) fluorescence (BDF25GR, Thorlabs). Post-objective laser power was measured with a power meter calibrated at 920 nm. The system was designed and built in collaboration with Thorlabs and is now a commercially available microscope system (Bergamo® II series). Then, 0.2-μm-diameter yellow-green fluorescent beads (FluoSpheres®, Thermo Fisher Scientific) attached to a glass slide were imaged using both the Gaussian and Bessel configurations of the microscope to measure the lateral and axial PSFs. 3D resolution was measured by taking the FWHM of the PSFs. The largest FOV achieved with the 16×/0.8 NA objective was 1.4 mm × 1.4 mm.

### Mouse surgical preparation. 
All animal experiments were conducted according to the National Institutes of Health guidelines for animal research. Procedures and protocols on mice were approved by the Animal Care and Use Committee at the University of California, Berkeley. In vivo imaging data in this study were collected from four wild-type (Jackson Laboratories, Black 6, stock no. 000664), one Thy1-GFP (neuronal GFP expression, Jackson Laboratories, Tg(Thy1-EGFP)MJrs/J, stock no. 007788), and two Aldh1l1-GFP (pan-glial GFP expression, Mutant Mouse Resource & Research Centers, Tg(Aldh1l1-EGFP)OFC789Gsat/Mmucd, stock no. 011015-UCD) mice socially housed under normal light cycle and room temperature. Cranial window implantation procedure has been described previously[72]. In brief, mice aged 3–4 months were anesthetized with 1–2% isoflurane in O$_2$ combined with the analgesic buprenorphine (SC, 0.1 mg/kg) and head-fixed in a stereotaxic apparatus (Kopf Instruments). A 3.5-mm-diameter craniotomy was made over the left V1 centered at −2.5 mm medial-lateral and 1 mm anterior–posterior to lambda. A glass window made of a single coverslip (Fisher Scientific, no. 1.5) was embedded in the craniotomy, flush with the skull, and sealed by a tissue adhesive (VetBond, 3M). A stainless-steel head-bar was then firmly attached to the skull with dental acrylic. Implanted mice were provided with a post-operative analgesia Meloxicam (SC, 5 mg/kg) for 2 days and allowed to recover for at least 2 weeks prior to imaging experiments.

### In vivo imaging. 
All imaging experiments were performed on head-fixed, awake mice. Prior to imaging, animals were briefly anesthetized with isoflurane and retro-orbitally injected with 50 μL of 5% (w/v) 70-kDa dextran-conjugated Texas Red fluorescent dye. Mice were then head-fixed under the objective lens. First, structural image stacks were taken using both Gaussian and Bessel beam paths at the largest FOV. For the two Aldh1l1-GFP transgenic mice, blood vessel (red) and glial (green) structures were imaged concurrently. Next, volumetric imaging of blood vessel dynamics was performed at 15 Hz (1024 × 1024-pixel frames) or 99 Hz (128 × 128-pixel frames) in distinct FOVs using the Bessel focus module. During some experiments, we concurrently imaged the mouse's ipsilateral eye illuminated by infrared LEDs using a camera (Mako U-130B) with an infrared filter. The bright spot on the upper left quadrant of the pupil (Fig. 4c) was the reflection of the infrared illuminator and did not affect pupil diameter analysis. For high-resolution capillary imaging, small FOVs were imaged at 30 Hz (512 × 512-pixels per frame) with 0.2 μm pixel size, first with Gaussian then Bessel foci. Due to the higher fraction of energy distributed in the side rings of a Bessel focus compared to a Gaussian focus[34], higher post-objective laser power was used for Bessel compared to Gaussian imaging at the same cortical depth (e.g., at 100 μm below pia surface, Bessel: 167–217 mW, Gaussian: 35–47 mW).

### Data analysis. 
All image processing, visualization, and analysis were performed in ImageJ and MATLAB® (MathWorks). Image sequences from Bessel functional data were registered to remove rigid lateral motion artifacts before further analysis. In all figures, superficial Bessel data (0–100 μm below surface of dura mater), unless otherwise stated, were visualized using the gray lookup table on the square root of fluorescence signal (normalized from minimum to maximum) to improve the visibility of dim structures without saturating bright structures. All Gaussian data

and all other Bessel data were visualized using the gray lookup table on their fluorescence signal (normalized from minimum to maximum).

For Supplementary Fig. 1 and its associated analysis, fluorescence signal and blood vessel size of 60 vessel segments were measured in both the Bessel image (900 frame average) and the corresponding Gaussian stack (5 frame average, 1 μm step size). For the Bessel image, line segments of 1-pixel thickness were drawn perpendicular to the blood vessels. Fluorescence signals were chosen to be the brightest pixel along the line segments and vessel sizes were the total numbers of pixels above a common threshold. The threshold was calculated by first selecting a region without vessels near the center of the image then adding 3 standard deviations to the mean pixel value of the region. For the Gaussian image stack, a line segment was drawn perpendicular to the same blood vessel in the Bessel image and the stack was resliced to show the axial cross-section of the vessel: fluorescence intensity was chosen to be the brightest pixel in this 2D cross-section, and size was represented by the FWHM at the widest point in the cross-section.

For Fig. 3, 1-min-long 15 Hz functional data were used to study the relationship between fluorescence intensity and vasodilation. First, we performed a 5-frame (0.33 s) moving average of the raw frames to remove high-frequency fluorescence signal variations due to blood flow. For four blood vessels, we extracted the time traces of the average signal from 256-pixel-area (450 μm$^2$) ROIs fully encompassed within the lumen of the vessels and calculated their ΔF/F traces with F being the median value of the time trace. Blood vessel diameters were measured by thresholding the moving-averaged images at 3 standard deviations above the mean background signal near the blood vessel of interest, and then calculating the number of pixels above threshold within a 10-pixel-wide line segment drawn perpendicular to the vessel. The same analysis procedures were performed on four capillaries in higher resolution (0.2 μm × 0.2 μm pixel size) Gaussian and Bessel data in Supplementary Fig. 2, with two additional steps. (1) Fluctuating background signal was removed from Bessel data with frame-by-frame subtraction of the background signal near the capillary of interest, then adding back a session-averaged background value. By doing this, capillary fluorescence measurements were decoupled from fluctuations originating from out-of-focus fluorescence such as large surface vessel dilations. (2) 30-Hz Gaussian and Bessel data were first temporally binned to 1 s resolution to suppress high-frequency blood-flow-introduced signal variation, followed by a 5-frame moving average on fluorescence and diameter measurements, which resulted in 5-s moving averages. For the signal correlation analysis in Fig. 3f and g, large volume (1.4 mm × 1.4 mm × 0.1 mm) Bessel TPLSM raw images were spatially binned to 64 × 64 ROIs (each 16 × 16 pixels) by averaging all pixels within a ROI. The time trace of each ROI was then extracted and correlated with the time trace of a specified target ROI with a sliding window of ±10 frames. The absolute maximum correlation value within the sliding window was used to represent the cross-correlation coefficient between the ROI and target ROI. Supplementary Fig. 3c was created by first dividing standard deviation of the image time stack by the average image and plotted on a color scale. Then, the dimmest 80% of pixels in the average image that represented non-vasculature tissue were set to black. Supplementary Fig. 3d was created in the same way, with an initial binning to 64 × 64 ROIs.

For the pupil-vessel entrainment experiment in Fig. 4, 10-min-long 15 Hz functional images at 1024 × 1024-pixel resolution were also binned to 64 × 64 ROIs (each 16 × 16 pixels) by averaging all pixels within an ROI. We used a threshold and fitting method adapted from Diego Barragan (Title: Tracking pupil using image processing, MATLAB Central File Exchange) to extract pupil diameter from concurrently recorded mouse pupil images. We fit the pupil in each video frame as an oval, and used its width, rather than height, as pupil diameter to reduce squinting and blinking artifacts. The time trace of each ROI was correlated with the mouse's pupil diameter with a sliding window of ±10 frames. The absolute maximum correlation value within the sliding window was used to represent the cross-correlation coefficient between the ROI and pupil diameter.

For 3D blood flow speed measurement presented in Fig. 5, blood vessel segments were hand-traced over the average intensity projection of 1-min-long 99 Hz datasets (two 208 μm × 208 μm × 80 μm volumes 0–80 μm below top pia surface in two mice; two 416 μm × 416 μm × 80 μm volumes 0–80 μm and 100–180 μm below the top pia surface, respectively, in two mice). Raw Bessel images were then resliced along the length of blood vessel segments and plotted in time as kymographs. Because RBCs excluded fluorescent dye and appeared as dark shadows, they generated diagonal dark streaks in kymographs[21]. We rejected from analysis blood vessel kymographs with little to no RBC streaks visible above the background fluorescence. We measured the 3D lengths of blood vessel segments from skeletonized Gaussian structural data and then nonlinearly transformed the horizontal axes of the kymographs from 2D Bessel projection length to 3D Gaussian structural length for further analysis. We adapted an automated method developed by Chhatbar and Kara[43] to measure the slope of RBC streaks using Sobel filtering and iterative Radon transforms and calculated blood flow speed every 50-frame (0.5 s) kymograph segment. We further rejected from analysis blood vessel segments that had more than one-third of its time trace showing very high (>5 mm/s) or reversed (<0 mm/s) flow speeds. Sizes of blood vessel segments were measured by taking the FWHM of a representative hand-drawn line intensity cross-section of each segment from the average intensity projection of the Bessel dataset.

**Reporting summary**. Further information on research design is available in the Nature Research Reporting Summary linked to this article.

## Data availability

Raw image sequences used to generate Figs. 2, 3, 5, and Supplementary Figs. 1, 3 are available on https://figshare.com/projects/Bessel_Vessel/89135. Raw data used to generate Fig. 4 and Supplementary Fig. 2 have large file sizes (>5 GB) and are available upon reasonable request to the corresponding author.

## Code availability

Custom skeletonization, blood vessel tracing, and pupil correlation code in MATLAB are available on https://figshare.com/projects/Bessel_Vessel/89135.

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

## Acknowledgements

We would like to acknowledge help from Srigokul Upadhyayula on developing the automated skeletonization code for structural analysis of vasculature, help from Qinrong Zhang on animal surgery, and helpful discussion from William Liberti. This work was sponsored by the US National Institutes of Health (J.L.F., N.J.: U01NS103489), the US Department of Energy (J.R., Lawrence Berkeley National Laboratory LDRD 20-116), and the Weill Neurohub.

## Author contributions

N.J. conceived of and supervised the project; J.R., W.S., J.P., and S.R. designed the microscope system; J.R., W.S., J.P., H.H., and J.L.F. built the system; J.L.F. and N.J. designed the experiments; J.L.F. and J.R. collected the data; J.L.F. and J.R. analyzed the data; J.L.F. and N.J. wrote the manuscript with input from all authors.

## Competing interests

Bessel focus scanning (N.J.) intellectual property has been licensed to Thorlabs, Inc. by Howard Hughes Medical Institute. W.S., J.P., and H.H. are full-time employees at Thorlabs, Inc. S.R. was a full-time employee at Thorlabs, Inc. for the duration of this study. The remaining authors (J.L.F., J.R.) declare no competing interests.
