## [Peer Review File · Nature Communications]

REVIEWER COMMENTS

Reviewer #1 (Remarks to the Author):

The manuscript by Fang et al. titled "High-speed volumetric two-photon fluorescence imaging of neurovascular dynamics" demonstrates application of Bessel 2P microscopy to structural and functional imaging of cortical microvasculature. This work is a collaboration of the US Berkeley investigators who previously have developed this imaging modality with their industrial partner using their new commercially available device. They describe the device and illustrate its capabilities for fast volumetric imaging of vascular dynamics and quantification of vasodilation and blood flow in awake mice with chronically installed optical windows. Overall, this is a striking demonstration of this novel application for the Bessel 2P technology that has a potential to exert a real impact in the field of cerebrovascular physiology and beyond. I'd like to congratulate the authors on this impressive achievement. There are a number of issues, however, both technological and biological, that should be clarified and expanded upon as detailed below. These comments are listed in the order they appear in the text, the order does not reflect their relative importance.

1. "Bessel TPLSM correlates fluorescence signal with vessel size" – this phrase, which appears multiple times in the manuscript, is awkward and requires rephrasing. In general, I recommend language editing by a native English speaker.
2. Please, clarify how simultaneous Gaussian/Bessel imaging was achieved, e.g., in Fig. 3. According to Fig. 1, it seems that the mirror would need to be switched in and out of the path.
3. It is not clear to me that the diameter measurements based on fluorescence intensity would work for capillaries. Capillaries would correspond to <6 microns vessels in Suppl. Fig. 1C. Can we see a zoom in of the relevant portion of this plot? If you limit the correlation analysis to this range of diameters, there may be little or no correlation. This is not a crime, but we need to be clear about this limitation.
4. An additional confounding factor for extracting diameters from intensity in small vessels with slow flow is movement of blood cells, red and white. This is for the same reason that allows measuring flow – the cells are not labeled. So, if cells move slowly in a small vessel, one by one or in clusters, the intensity would fluctuate even in the absence of a real diameter change. I think that this is the reason for low correlation in Fig. 3D ROI#1 – this is a small dural arteriole that probably has slow flow. Other ROIs in this figure are arterioles on the brain surface that dilate robustly and have very high flow, such that movement of cells does not affect the intensity measurement.
5. My point (4) is a major confounding factor in Suppl. Fig. 2. In addition, (i) the resolution of standard (Gaussian) 2P microscopy is borderline for measurements of capillary diameters, and (ii) the Gaussian diameter measurements rely on detection of the edge (vessel wall) that gets compromised when a dark cell is squeezing through. So, this figure is comparing 2 questionable capillary diameter measurements that makes me question the conclusion about the presence of capillary dilation. Capillary dilation also happens to be a highly debated issue, and many of the involved biologists would not be aware of these technological pitfalls. So, this is a dangerous figure and conclusion to throw out there is the community.
6. In general, with Bessel-2P, larger and fast flowing vessels will be good candidates for measuring diameters and capillaries – for measuring velocities. This is not unlike the conventional 2P, but Bessel-2P of course has a critical advantage of high throughput.
7. Fig. 3F-G. These "networks" show surface arteries (red in F) and surface veins (red in G). They should not be called networks, because circulation connects arteries to veins through capillaries, i.e., the same network. Can we see the actual timecourse of the signal from these arteries and veins superimposed? How big is the change in the veins relative to arteries? Suppl. videos clearly show dilation in surface arteries but not veins. Veins as a matter of fact cannot actively dilate (or constrict), they simply don't have muscles to do that. However, active dilation of arteries would increase pressure in the tissue leading to partial collapse of veins – maybe this is what you are seeing here. I bet that the venous change is very small.
8. Fig. 4 and the associated video: ROI#1 is a surface artery, ROI#2 (the wiggly vessel at the

bottom right) is a dural arteriole. It makes sense that they are not dilating in unison, because they are regulated by different mechanisms.

9. Please clarify the depth range of the measurements. Different numbers are present in different places incl. 100, 80 and 180 microns. Which one is it? What was the actual Bessel beam extent in z?

10. Was any data rejected due to motion, e.g., during grooming? What was the rejection criterion?

11. Line 238 "...allowed us to measure blood flow in penetrating blood vessels and capillaries" – there has to be an angle between the Bessel beam and the vessel to measure velocity. Please, clarify.

12. Line 241 "...vessels captured with Bessel focus scanning do not suffer from axial motion artifacts" – true for diameters, not velocities.

13. There is a missing paragraph in Discussion that has to do with limitations. This should include a number of issues raised above, higher heating compared to conventional 2P, the requirement of sparse labeling, dependence on the angle between the Bessel beam and the vessel, etc.

14. Line 266: "Large surface vessels aberrated the Gaussian focus and led to reduced excitation efficiency" – is it aberration, absorption or both?

15. Line 276: "We also observed size changes in capillaries" – See (4) and (5). When these technological limitations are eventually mitigated, it may turn that capillaries do not in fact dilate (and there are some good reasons to believe that they don't).

16. Line 285: "...neural activity dynamics that modulates the local metabolic demands of the brain tissue" – this view was popular some time ago, but now many believe that vasodilation is regulated feed forward by vasoactive neurotransmitters, not by a feedback from how much energy has been consumed. E.g., ACh is a vasodilator <https://pubmed.ncbi.nlm.nih.gov/16467392/> that can explain these very nice data that you show in Fig. 4 and the corresponding video.

Reviewer #2 (Remarks to the Author):

The manuscript by Fan JL et al. title "High-speed volumetric two-photon fluorescence imaging of neurovascular dynamics" is an interesting technical paper showing how to apply Bessel 2P beams to capture pseudo 3D images of the neocortex in vivo. In conventional 2P, a diffraction limited laser spot is produced to generate fluorescence, giving inherent optical sectioning. Bessel optics instead create a "lance" like excitation process (skinny in x&y yet very long in the z). Here a single "2D" image captures a volume all at once. This paper represents the first use of Bessel beams for cerebral microvascular research in vivo (though other in vivo vascular beds have been imaged with Bessel beams before PMID 24876996) and in this sense the paper is strong. However, my overall enthusiasm is limited for a publication in Nature Communication, as this is more of a technical paper with very little novel biology. True that vessel diameter changes and RBC velocity changes can be captured faster or more abundantly, but there are no large insights or discoveries created from this work. One hint of this is that the authors observed intriguing correlated and anti-correlated diameter changes to pupil diameter. However, this phenomenon isn't explored at all and while a very interesting measurement, ideally for Nat Comms publication the entire paper should be about understanding this. I like this paper for its technical strengths and the advances that it can offer others in its application, but in my opinion it is better suited for a more specialized, perhaps technical journal.

High-speed volumetric two-photon fluorescence imaging of neurovascular dynamics

Author response:

We thank both reviewers for their helpful comments and suggestions, which have helped us greatly improve the manuscript. We have made point-by-point responses below and have indicated where edits were made in the revised manuscript. We hope that the reviewers find our responses satisfactory and consider the manuscript acceptable for publication in Nature Communications.

Reviewer 1:

The manuscript by Fang et al. titled “High-speed volumetric two-photon fluorescence imaging of neurovascular dynamics” demonstrates application of Bessel 2P microscopy to structural and functional imaging of cortical microvasculature. This work is a collaboration of the US Berkeley investigators who previously have developed this imaging modality with their industrial partner using their new commercially available device. They describe the device and illustrate its capabilities for fast volumetric imaging of vascular dynamics and quantification of vasodilation and blood flow in awake mice with chronically installed optical windows. Overall, this is a striking demonstration of this novel application for the Bessel 2P technology that has a potential to exert a real impact in the field of cerebrovascular physiology and beyond. I’d like to congratulate the authors on this impressive achievement. There are a number of issues, however, both technological and biological, that should be clarified and expanded upon as detailed below.

These comments are listed in the order they appear in the text, the order does not reflect their relative importance.

We appreciate the reviewer’s positive comments on our technology and confidence in its potential impact on the field of cerebrovascular physiology. Please find below our point-by-point responses. We hope we have adequately addressed all issues that have been brought up.

1. “Bessel TPLSM correlates fluorescence signal with vessel size” – this phrase, which appears multiple times in the manuscript, is awkward and requires rephrasing. In general, I recommend language editing by a native English speaker.

We thank the reviewer for bringing attention to the awkward phrasing. We updated parts of the manuscript where this phrase was used (Abstract lines 20-22, Introduction lines 76-77, and Discussion lines 260-262) to better convey our finding that fluorescence signal in Bessel TPLSM is proportional to vessel diameter.

2. Please, clarify how simultaneous Gaussian/Bessel imaging was achieved, e.g., in Fig. 3. According to Fig. 1, it seems that the mirror would need to be switched in and out of the path.

In Figs. 3B, C and Supplementary Fig. 1, we compared the same blood vessels using Gaussian and Bessel imaging. These images were not acquired simultaneously. When imaging our animals, we first took a

Gaussian 2P z-stack of a FOV, followed by Bessel 2P of the same FOV. Data comparing the two imaging modalities in the manuscript were therefore taken at different times, but always within the same imaging session. To make this point more explicit, we have included “captured at different times” in the Fig. 3B and Supplementary Fig. 1C captions.

3. It is not clear to me that the diameter measurements based on fluorescence intensity would work for capillaries. Capillaries would correspond to <6 microns vessels in Suppl. Fig. 1C. Can we see a zoom in of the relevant portion of this plot? If you limit the correlation analysis to this range of diameters, there may be little or no correlation. This is not a crime, but we need to be clear about this limitation.

We apologize for misleading the reviewer by mistakenly labeling Supplementary Fig. 1 as “Relationship between blood vessel diameter and fluorescence intensity”. The correct title should be “Relationship between blood vessel diameters measured with Gaussian versus Bessel TPLSM methods”. Therefore, Supplementary Fig. 1C is a comparison of blood vessel diameters as measured from Gaussian (Supplementary Fig. 1A) and Bessel (Supplementary Fig. 1B) images, in order to show that both imaging approaches gave rise to similar vessel sizes (because we acquired the Gaussian and Bessel images at different times, the dynamic vasodilation and constriction prevented exact correspondence between the vessel sizes measured from these images).

During revision, we collected more data on capillaries using smaller pixel sizes. Focusing on capillaries with diameters less than 6 μm , we still observed a strong correlation between the capillary diameters measured with Gaussian and Bessel TPLSM methods (Fig. R1).

Figure R1. Scatter plot of vessel diameters measured from Gaussian versus Bessel high-resolution data (0.2 μm pixel size).

Separately here, we would like to address the reviewer’s comment “It is not clear to me that the diameter measurements based on fluorescence intensity would work for capillaries”, which was most likely directed to the data in Fig. 3. In Fig. 3C, we showed that compared to Gaussian focus, the fluorescence signal of blood vessels measured by Bessel focus was more correlated with the vessel size. However, we did not intend to claim that one can extract the diameter value of a blood vessel from its fluorescence intensity in Bessel TPLSM alone.

In bottom panel of Fig. 3C, we plotted the Bessel fluorescence signal versus diameter for distinct vessel segments in the same field of view. Here, variations in local tissue scattering, aberrations and the orientation of blood vessels relative to the

excitation focus led to deviations away from the diagonal line. Indeed, it would be inadvisable to compare the signal of two vessels in order to reach conclusion on their relative sizes. However, for the same blood vessel segment, changes in its fluorescence signal do indicate changes in its vessel size, as we demonstrated in Fig. 3D (below in our response to reviewer point 5, we include additional discussion and data to support that this statement also holds true for capillaries). To make this point explicit, we have added to lines 157-163:

“The observed positive correlation between fluorescence brightness and diameter of distinct vessel segments in the Bessel image, although high in value, cannot be relied upon to reach conclusions on relative sizes for different vessels. Even for the vessels in the same field of view, variations in local tissue scattering, aberration, and orientation relative to the excitation focus led to deviations away from perfect signal-to-size correlation. However, for the same vessel segment, changes in its fluorescence brightness in Bessel TPLSM should be strongly correlated with changes in its size, and thus can be reliably used to detect vasodilation and vasoconstriction.”

4. An additional confounding factor for extracting diameters from intensity in small vessels with slow flow is movement of blood cells, red and white. This is for the same reason that allows measuring flow – the cells are not labeled. So, if cells move slowly in a small vessel, one by one or in clusters, the intensity would fluctuate even in the absence of a real diameter change.

We thank the reviewer for bringing up this important point on how blood flow can change fluorescence intensity in the absence of diameter change. As described in Methods (Lines 385-386 in the original manuscript, Lines 422-423 in the revised manuscript), we performed a 5-frame (0.33 second) moving average of the raw frames to remove high-frequency fluorescence signal variations due to blood flow. For the newly acquired capillary data (more information in our response to reviewer’s point 5), 5-sec moving average was applied (Lines 430-437). We also added the following in lines 180-186:

“For all example vessels, vessel diameter and fluorescence brightness were highly positively but not perfectly correlated. The lack of perfect correlation was caused by blood cells, which were not labeled by fluorescent dye. Whenever they flowed through the excitation focus, they reduced the fluorescence brightness even when the vessel size stayed constant. To reduce blood-cell-induced fluctuations in fluorescence, we temporally averaged the fluorescence and diameter data (Methods; 0.33 sec moving average for non-capillaries and 5 sec moving average for capillaries) and observed the fluorescence brightness to closely reflect the vessel diameter.”

I think that this is the reason for low correlation in Fig. 3D ROI#1 – this is a small dural arteriole that probably has slow flow. Other ROIs in this figure are arterioles on the brain surface that dilate robustly and have very high flow, such that movement of cells does not affect the intensity measurement.

Thanks to the reviewer’s close attention to Fig. 3 ROI#1, we noticed upon closer inspection of the raw data that there was a small vessel branching off the vessel segment chosen for ROI#1 (Fig. R2, 1, red arrow). The existence of this small blood vessel led to errors in the diameter measurement. In the revision, we chose a different segment of the same dural arteriole (Fig. R2, 2) and measured fluorescence and diameter changes. We found them to be much more closely correlated than before. We have replaced ROI#1 in the updated Fig. 3A, D, and E.

5. My point (4) is a major confounding factor in Suppl. Fig. 2. In addition, (i) the resolution of standard (Gaussian) 2P microscopy is borderline for measurements of capillary diameters, and (ii) the Gaussian diameter measurements rely on detection of the edge (vessel wall) that gets compromised when a dark cell is squeezing through. So, this figure is comparing 2 questionable capillary diameter measurements that makes me question the conclusion about the presence of capillary dilation. Capillary dilation also happens to be a highly debated issue, and many of the involved biologists would not be aware of these technological pitfalls. So, this is a dangerous figure and conclusion to throw out there in the community.

We thank the reviewer for raising these concerns about Supplementary Fig. 2 (see below), which we have completely revised with newly collected high spatial resolution capillary data (0.2 $\mu\text{m}/\text{pixel}$, 30 Hz rate; 2 mice, representative data from 1 mouse included in revised Supplementary Fig. 2). Given that capillary dilation is a highly debated issue in the research community, we want to be very careful about our data collection and analysis methods, as well as clear about the conclusions we can make in the manuscript.

Regarding point (i) on resolution of 2P microscopy being borderline for measurement of capillary diameters: Given that we are measuring diameter changes in well-isolated capillary segments rather than resolving structures (i.e., we are only localizing the edges of the blood vessel), we can achieve greater measurement accuracy (but not resolution) than the lateral resolution of the microscope ($\sim 0.65 \mu\text{m}$ for Bessel focus). In other words, we should be able to detect capillary diameter changes smaller than 0.65 μm .

Regarding point (ii) about temporal variation of fluorescence inside capillaries due to travelling dark cells affecting diameter measurements: We only performed diameter measurements on high-SNR capillaries after temporal averaging, and thus avoided the issue of having to measure vessel edges when the SNR is compromised due to a dark cell. (Also see our more detailed answer to reviewer's point 4.)

We performed Gaussian 2P imaging on a few capillary segments while simultaneously measuring pupil diameter (Supplementary Figs. 2A-C). Capillary diameters and fluorescence were measured on 30-frame-averaged (1s-averaged) images and further smoothed with a 5-second moving average. We observed

changes in diameter in some capillaries. Sometimes, capillary dilation reached almost 1 μm over several seconds (see Supplementary Fig. 2D for an example). Our findings are consistent with literature in 2 ways: only a fraction of capillaries dilate significantly in response to stimulation (Stefanovic 2008, Hall 2014), and dilating capillaries show a 5-20% change in diameter (Tian 2010, Cai 2018, Stefanovic 2008, Hall 2014). Capillary dilation may be passively caused by pressure changes in nearby connected arterioles, or result from an active mechanism involving pericytes, as suggested in Hall 2014 and Cai 2018.

We also performed Bessel 2P imaging of the same FOVs. Even though Bessel and Gaussian imaging were not simultaneous, we similarly observed diameter changes in the same capillaries (Supplementary Figs. 2C, E). Additionally, we saw strong correlation between capillary diameter and lumen fluorescence (Supplementary Fig. 2F), which was consistent with our findings in larger vessels (Fig. 3E). With this new high-resolution supplementary data, we are now confident that vessel dilation does appear in some capillaries and that the correlation between changes in Bessel fluorescence intensity and changes in the vessel size also exist for capillaries. We have added this information to lines 173-179 of the revised manuscript:

“We carried out similar analyses for capillaries (vessels with diameters less than 6 μm ; Supplementary Fig. 2) in 2 animals, at 30 Hz and 0.2 μm pixel size. In both Gaussian (Supplementary Fig. 2C, D) and Bessel (Supplementary Fig. 2E) modes, we measured diameters of capillary segments and observed dilation and constriction from some capillaries. Our observation that some but not all capillaries dilate is consistent with previous studies where only a fraction of capillaries was found to undergo measurable dilation (36, 37). Those capillaries that did dilate show similar magnitudes of diameter change to those observed previously in response to forepaw or whisker pad stimulations (36-39).”

6. In general, with Bessel-2P, larger and fast flowing vessels will be good candidates for measuring diameters and capillaries – for measuring velocities. This is not unlike the conventional 2P, but Bessel-2P of course has a critical advantage of high throughput.

We agree with the reviewer that one critical advantage of Bessel 2P is its high throughput. Hopefully our responses above and additional data on capillaries have also convinced the reviewer of the additional advantage of Bessel 2P in how changes in fluorescence signal are correlated with changes in size for the same blood vessel, which provides an alternative approach to detect vasodilation.

Supplementary Figure 2. Capillary dilation and constriction probed by Gaussian and Bessel TPLSM.

(A) A 1.4 mm x 1.4 mm x 0.1 mm volume of vasculature imaged with Gaussian TPLSM, color coded by depth. (B) High-resolution Gaussian and Bessel images of capillaries. For each area (within the white squares in A), Gaussian images were taken first, followed by Bessel images, both acquired at 30 Hz with 0.2 μm pixel size for 8 minutes. (C) Time traces of diameters of 4 capillary segments from Gaussian images in B, plotted with simultaneously measured pupil diameter. (D) Example pupil and 1-second averaged Gaussian images of Capillary 4 at two time points (indicated by yellow lines in C) showing changes in pupil and capillary diameters. (E) Time traces of fluorescence signal changes and diameters of the same 4 capillary segments from Bessel images in B, plotted with simultaneously measured pupil diameter. Unlike larger vessels whose size correlations with pupil diameter were time-invariant (Fig. 4D), capillaries had time-varying size correlations with pupil diameter over minutes. (F) Scatter plot of fluorescence versus capillary diameter data in E. Scale bars: A: 200 μm , B: 20 μm , D: Pupil: 1mm, Gaussian: 5 μm . Post-objective power: Gaussian: 17 mW, Bessel: 112 mW.

7. Fig. 3F-G. These “networks” show surface arteries (red in F) and surface veins (red in G). They should not be called networks, because circulation connects arteries to veins through capillaries, i.e., the same network. Can we see the actual timecourse of the signal from these arteries and veins superimposed? How big is the change in the veins relative to arteries? Suppl. videos clearly show dilation in surface arteries but not veins. Veins as a matter of fact cannot actively dilate (or constrict), they simply don’t have muscles to do that. However, active dilation of arteries would increase pressure in the tissue leading to partial collapse of veins – maybe this is what you are seeing here. I bet that the venous change is very small.

We thank the reviewer for bringing up this distinction. We have edited the manuscript to remove incorrect usage of “networks”.

Please see the new Supplementary Figure 3 for signals from ROIs corresponding to arteries and veins superimposed. Indeed, as predicted by the reviewer, the ROI representing the vein (blue trace) showed much smaller changes in Bessel fluorescence signal (thus vessel size) than the ROI representing the artery (red trace). We also carried out additional analysis of this dataset, where for pixels representing vessels, we plotted the ratio of the standard deviation versus the mean for the fluorescence time trace of each pixel (Supplementary Fig. 3C, D, updated Methods). Similarly, we observed larger changes in fluorescence signal (thus vessel size) in arteries than veins. We have added text to the Results section lines 196-201 to describe this finding:

“The ROI in Fig. 3G had a $\Delta F/F$ time trace with a standard deviation 3.6× greater than the ROI in Fig. 3F (Supplementary Fig. 3). Therefore, the ROI in Fig. 3G likely belonged to surface arterioles and the ROI in Fig. 3F likely belonged to venules due to the differing dilatory mechanisms between arterioles and venules (3, 19, 41). Pixels with positive correlation with the two ROIs in Figs. 3F and 3G, respectively, revealed surface venule and arteriole populations.”

8. Fig. 4 and the associated video: ROI#1 is a surface artery, ROI#2 (the wiggly vessel at the bottom right) is a dural arteriole. It makes sense that they are not dilating in unison, because they are regulated by different mechanisms.

We thank the reviewer for providing context to our findings. We have added a sentence “Based on their location and morphology, ROI 1 and ROI 2 were identified to be a surface artery and a dural arteriole, respectively, with distinct mechanisms regulating their dilation (41).” to the Results section lines 219-220 to mention this difference.

9. Please clarify the depth range of the measurements. Different numbers are present in different places incl. 100, 80 and 180 microns. Which one is it? What was the actual Bessel beam extent in z?

We thank the reviewer for bringing up this point. The Bessel focus had a FWHM of 67 μ m. Depending on the labeling strength and the axial position of the Bessel focus, it covered a volume of \sim 100 μ m thickness. When the Bessel focus is placed at superficial depth, the top edge of the Bessel focus sometimes fell outside the brain, which led to a probe volume with a thickness less than 100 μ m (e.g., 80 μ m). The 180 μ m number that the reviewer referred to was most likely from the caption for Fig. 5G, where we showed the aggregated data from 4 mice, including 63 vessel segments 0-180 μ m below dura. In other words, these data were not collected simultaneously from a 180- μ m-thick brain volume but from separate \sim 80 or \sim 100- μ m-thick brain volumes at different depths. We have rephrased the figure caption to make it more obvious that these data were from 4 different mice.

10. Was any data rejected due to motion, e.g., during grooming? What was the rejection criterion?

All brain imaging data was included in analysis without any pre-selection or rejection. (As described in Methods\Data analysis, images were registered to remove rigid lateral motion artifacts. With our surgical preparation, the resistance of Bessel foci against axial motion artifacts prevented any axial motion artifacts.) The only data not included were the pupil imaging data in some sessions in which the animal squinted its eyes excessively, making pupil size measurement difficult.

11. Line 238 “..allowed us to measure blood flow in penetrating blood vessels and capillaries” – there has to be an angle between the Bessel beam and the vessel to measure velocity. Please, clarify.

It is indeed correct that there needs to be an angle between the Bessel beam and the vessel to measure velocity. We have rephrased the relevant discussion paragraph (lines 265-271) to make this point more explicit:

“By extending the excitation volume axially, a Bessel focus allowed us to measure blood flow in vessels and capillaries that were not parallel to the axis of the Bessel focus. In contrast, in conventional TPLSM imaging with the more axially confined Gaussian focus, many vessels appear only as cross sections, which makes blood flow measurement difficult. If it is important to measure blood flow from vessels that are parallel with the axis of the Bessel focus, the annular illumination at the back focal plane of the microscope objective can be translated to generate another Bessel focus travelling along a different direction from the original focus and a distinct projected view of the vasculature.”

12. Line 241 “..vessels captured with Bessel focus scanning do not suffer from axial motion artifacts” – true for diameters, not velocities.

We believe that velocity measurements with Bessel focus indeed do not suffer from axial motion artifacts. With Bessel focus scanning, the resulting images are the projection images of the volume along the axial direction. We measured blood flow velocity by tracking blood cell movement in the projected lateral plane. Because the projected images in the lateral plane are resistant to axial motion artifacts, the resulting velocity measurements are also resistant to axial motion artifacts.

13. There is a missing paragraph in Discussion that has to do with limitations. This should include a number of issues raised above, higher heating compared to conventional 2P, the requirement of sparse labeling, dependence on the angle between the Bessel beam and the vessel, etc.

We thank the reviewer for this suggestion. We have addressed the issue regarding the dependence on the angle between Bessel focus and vessel in lines 265-271 (see reviewer point 11).

We added a discussion on higher heating in lines 288-292:

“To image vasculature structures at depth, more average power needs to be deposited into the mouse brain with Bessel TPLSM than Gaussian TPLSM. However, the highest average power utilized in this study (up to 217 mW) was under the threshold for heating-induced damage in the mouse brain *in vivo* (47) and below the average powers that were utilized in other high-speed brain imaging technologies (48, 49).”

We have a discussion on the requirement of sparse labeling in lines 280-283:

“Because structures at different depths within the excitation volume appear in the same projected images, Bessel TPLSM performs best with sparsely labeled and high-contrast structures, making it suitable for imaging neurovasculature, which are sparse in the brain and typically labeled with bright fluorescent dyes.”

14. Line 266: “Large surface vessels aberrated the Gaussian focus and led to reduced excitation efficiency” – is it aberration, absorption or both?

We thank the reviewer for pointing out the possible role of absorption. Indeed, both aberrations and absorption play a part in reducing excitation efficiency. For the 920nm excitation light and red fluorescence emission, blood does not have high absorption, and it is primarily aberrations that reduced excitation efficiency.

15. Line 276: “We also observed size changes in capillaries” – See (4) and (5). When these technological limitations are eventually mitigated, it may turn that capillaries do not in fact dilate (and there are some good reasons to believe that they don’t).

Please see our detailed responses to reviewer’s points 4 and 5. Consistent with some previous publications, we indeed observed dilation in some capillaries. We have rephrased this sentence into “We also observed size changes in some capillaries” to be more precise.

16. Line 285: “..neural activity dynamics that modulates the local metabolic demands of the brain tissue” – this view was popular some time ago, but now many believe that vasodilation is regulated feed forward by vasoactive neurotransmitters, not by a feedback from how much energy has been consumed. E.g., ACh is a vasodilator <https://pubmed.ncbi.nlm.nih.gov/16467392/> that can explain these very nice data that you show in Fig. 4 and the corresponding video.

We thank the reviewer for this information. We have edited the sentence to remove the outdated viewpoint and cited the paper that the reviewer referred to in Lines 318-320: “Since pupil diameter is closely tied to brain arousal levels (62-64) and cortical states (42, 65), this entrainment of vessel and pupil dilations reflects underlying neural activity dynamics including the release of vasoactive neurotransmitters (66).”

Reviewer #2 (Remarks to the Author):

The manuscript by Fan JL et al. title "High-speed volumetric two-photon fluorescence imaging of neurovascular dynamics" is an interesting technical paper showing how to apply Bessel 2P beams to capture pseudo 3D images of the neocortex in vivo. In conventional 2P, a diffraction limited laser spot is produced to generate fluorescence, giving inherent optical sectioning. Bessel optics instead create a "lance" like excitation process (skinny in x&y yet very long in the z). Here a single "2D" image captures a volume all at once. This paper represents the first use of Bessel beams for cerebral microvascular research in vivo (though other in vivo vascular beds have been imaged with Bessel beams before PMID 24876996) and in this sense the paper is strong. However, my overall enthusiasm is limited for a publication in Nature Communication, as this is more of a technical paper with very little novel biology. True that vessel diameter changes and RBC velocity changes can be captured faster or more abundantly, but there are no large insights or discoveries created from this work. One hint of this is that the authors observed intriguing correlated and anti-correlated diameter changes to pupil diameter. However, this phenomenon isn't explored at all and while a very interesting measurement, ideally for Nat Comms publication the entire paper should be about understanding this. I like this paper for its technical strengths and the advances that it can offer others in its application, but in my opinion it is better suited for a more specialized, perhaps technical journal.

We thank the reviewer for the appreciation of the technical strength and impact of our method. We respectfully disagree that understanding the underlying biological mechanism of novel biological discoveries (e.g., the coupling of vessel dilation to pupil diameter) is essential and has to be included in this paper, which has been written as a technical paper with extensive biological applications and demonstrations of its ability for new biological discoveries.

EVIEWS' COMMENTS

Reviewer #1 (Remarks to the Author):

The authors sufficiently addressed all my concerns. This was a strong paper to start with, the revised version is stellar.